# Multistability and anomalies in oscillator models of lossy power grids

Robin Delabays [1] ✉, Saber Jafarpour [1,2] & Francesco Bullo [1]

The analysis of dissipatively coupled oscillators is challenging and highly relevant in power grids. Standard mathematical methods are not applicable, due to the lack of network symmetry induced by dissipative couplings. Here we demonstrate a close correspondence between stable synchronous states in dissipatively coupled oscillators, and the winding partition of their state space, a geometric notion induced by the network topology. Leveraging this winding partition, we accompany this article with an algorithms to compute all synchronous solutions of complex networks of dissipatively coupled oscillators. These geometric and computational tools allow us to identify anomalous behaviors of lossy networked systems. Counterintuitively, we show that loop flows and dissipation can increase the system's transfer capacity, and that dissipation can promote multistability. We apply our geometric framework to compute power flows on the IEEE RTS-96 test system, where we identify two high voltage solutions with distinct loop flows.

## Synchronization and flow networks

The history of scientific investigation about synchronization is traditionally traced back to Huygens' observation of an "odd kind of sympathy" in the XVIIth century[1]. It is, however only in the last decades that a tractable framework has been developed[2–4], thanks in particular to the pioneering works of Winfree in the 1960s[5], and Kuramoto in the 1970s–80s[6,7]. Shortly thereafter, the problem of synchronization has been embedded in the framework of network systems[8–10], first based mostly on numerical simulations, evolving progressively towards more and more analytical results[2,4,11]. Even in the simplest form of coupled oscillators, the interplay between dynamics and network structures leads to rich and sometimes unexpected behaviors.

The interactions between synchronizing oscillators is naturally interpreted as a flow of information or commodity between the nodes of a network. This dual interpretation of synchronization and flows is predominant in the modeling of voltage dynamics in high voltage power grids[12,13]. Indeed, in power systems, a rotating turbine in a plant accumulates kinetic energy and accelerates if all the power it produces is not transmitted to its neighboring buses. While there is a natural link between synchronization and flow balance in power grids, a similar duality underlies dynamical systems as diverse as spring-connected

rotating masses[11], motion planning[14,15], or chemical oscillations[7] to name but a few. The rate of change of an oscillator's state is then determined by the imbalance of the flows received from or sent to its neighbors. When the flows of commodity balance out at each agent, such that all agents have identical rates of change, then the relative positions of the agents are constant in time: we say that they are synchronized. In particular, this is the desired state for an AC power grid.

## Lossless oscillator networks

One of the simplest models of synchronization considers a set of oscillators, each described by a phase $\theta \in \mathbb{S}^1 \simeq [-\pi, \pi)$, interacting with each other through a $2\pi$-periodic coupling, function of their phase difference,

$$m_i \ddot{\theta}_i + d_i \dot{\theta}_i = \omega_i - \sum_{j=1}^{n} a_{ij} f_{ij}(\theta_i - \theta_j), \quad (1)$$

for $i \in \{1, \ldots, n\}$. The real parameters $m_i$ and $d_i$ represent the effective inertia and damping of oscillator $i$ respectively, and $\omega_i$ is the natural frequencies that the oscillator would hold without interactions. The function $f_{ij}$ is the coupling between nodes $i$ and $j$ and $a_{ij} \in \mathbb{R}_{\geq 0}$ is the edge weight that scales the strength of the coupling and determines

[1]Center for Control, Dynamical Systems, and Computation, UC Santa Barbara, Santa Barbara, CA 93106-5070, USA. [2]School of Electrical and Computer Engineering, Georgia Institute of Technology, Atlanta, GA 30332-0250, USA. ✉e-mail: robindelabays@ucsb.edu

the underlying network structure. These coefficients are nonzero if and only if oscillators $i$ and $j$ interact. The system described by equation (1) evolves on the $n$-torus $\mathbb{T}^n = (\mathbb{S}^1)^n$. The oscillators are frequency synchronized (or phase-locked) if, at some point in time, $\dot{\theta}_i \equiv \varphi$, for all $i$. The intrinsic compact nature of each oscillator's domain and the continuity of the coupling function require $f_{ij}$ to be nonlinear, and periodic in the systems of interest here. Whereas linear networked systems are well-understood[16], the nonlinear nature of the coupling between oscillators can lead to rich and intricate behaviors[13,17].

The vast majority of the literature about the synchronization of coupled oscillators assumes symmetric couplings, i.e., two coupled oscillators influence each other with the same strength. In the flow network interpretation, symmetric couplings correspond to lossless flows, i.e., the flow of commodity between $i$ and $j$ contributes with equal magnitude and opposite sign to each end of the edge between $i$ and $j$. Shortly put in mathematical terms, $f_{ij}(x) = -f_{ji}(-x)$. Consequences of this strong relation between $f_{ij}$ and $f_{ji}$ are in particular: (i) conservation of the total flow in the system, simplifying the calculation of the asymptotics of equation (1) and (ii) symmetry of the Jacobian matrix of the system, guaranteeing nice and convenient spectral features. The properties of systems with lossless couplings allowed to derive a long list of results about their dynamics: conditions for existence and uniqueness of their synchronous states[11,18,19]; multistability[13,20]; and clustering[21,22] to name but a few. An approach, common to various works, is to design a fixed-point iteration[18–20], whose convergence is guaranteed under some convexity properties of the energy landscape of the system[23,24].

## Challenges in the lossy oscillator systems

While the lossless assumption is reasonable in many cases, it is often not realistic and can lead to inaccurate predictions (see the power flow problem in the Results Section). In the flow interpretation, the transfer of a commodity, say electric power, is subject to dissipation, e.g., due to line resistance, meaning that the amount sent from $i$ to $j$ is strictly larger than the amount received by $j$ from $i$. In mathematical terms, dissipation are introduced in equation (1) by adding a term to the coupling function

$$m_i\ddot{\theta}_i + d_i\dot{\theta}_i = \omega_i - \sum_{j=1}^{n} a_{ij}\left[ f_{ij}(\theta_i - \theta_j) + g_{ij}(\theta_i - \theta_j) \right], \quad (2)$$

satisfying $g_{ij}(x) = g_{ji}(-x)$. Note that any pair of coupling functions (from $i$ to $j$ and from $j$ to $i$) can be decomposed as the sum of $f_{ij} + g_{ij}$ with the above properties [see equations (38) and (39) in the Methods Section].

The importance of understanding the more realistic case of dissipative couplings motivated the early work by Sakaguchi and Kuramoto[25,26] and is still an active field of research. Recent numerical investigations[27,28] as well as analytical studies in regular systems[29–31] are beginning to shed light on a more in-depth understanding of dissipative networks. More generally, the extension of standard approaches to more realistic systems is gaining momentum in the fields of synchronization and complex networks[32–34].

Up to this day, it is unclear to what extent the properties enjoyed by lossless networks are preserved in more realistic, dissipative systems. In the global scientific aim of faithful modeling of real systems, it is of utmost importance to decipher the impact of dissipation in standard models of networked dynamics. Indeed, conditions for existence, uniqueness, and multiplicity of synchronous states or for the emergence of clustering in lossless systems[11,13,20–22] are yet to be adapted to their dissipative counterpart. Furthermore, it is now largely documented[35] that phase frustration can lead to the occurrence of solitary and chimera states[28,36–40], that are extensively studied, but still only partially understood.

Understanding dissipative systems is challenging for a number of reasons. In such systems, flow conservation is lost and the linearization of the system typically loses its symmetry. Furthermore, while equation (1) can be formalized as a gradient system over an energy landscape, this property immediately fails in dissipative systems such as equation (2). Therefore, technical approaches based on energy landscapes are not applicable any longer. Incorporating dissipation in the system even requires to re-think the intuitive vectorial formulation of equation (1), in order to recognize the directionality of flows. Notice that, surprisingly, even a clear vectorial form of the dissipative dynamics is lacking in the literature.

## Models of lossy power grids

In the context of power grids, taking

$$f_{ij}(x) = \sin(x), \quad (3)$$

in equation (1) yields precisely the lossless approximation of the swing equations[12,13], describing the time evolution of voltage phase angles, with $\omega_i$ being the power injection or consumption at bus $i$. In normal operation, it is desired that the system is maintained in the vicinity of a synchronous state of the swing equations, which is reached at the solution of the lossless power flow equations (see the Methods Section).

As discussed above, while the lossless approximation can be fair and useful in power grids modeling, it is never exact. Therefore, accurate mathematical analysis of voltage dynamics and power flows requires to take dissipation into consideration. Considering resistive losses in the swing equations boils down to taking (see Methods Section)

$$f_{ij}(x) = \cos(\phi_{ij})\sin(x), \quad g_{ij}(x) = \sin(\phi_{ij})[1 - \cos(x)]. \quad (4)$$

The mathematical challenges encountered when relaxing the lossless line assumption confined most of the literature to numerical investigations[27,28].

## Objectives and contributions

The overarching goal of this article is to develop an analytical framework to characterize the location, properties, and stability of synchronous solutions of dissipative oscillator networks. In the task of globally characterizing synchronous states of lossless oscillator networks, an instructive and effective approach has been to leverage the concepts of winding numbers and winding cells[20]. Given a cycle of oscillators $\sigma = (\theta_1, \ldots, \theta_{|\sigma|}, \theta_1)$, the associated winding number $q_\sigma(\boldsymbol{\theta})$ counts the number of times the oscillators' angles wrap around the origin when following $\sigma$ (a rigorous definition is given in the Results Section). A winding cell is a subset of the $n$-torus $\mathbb{T}^n$ whose points share the same winding number around each cycle. Winding cells form a partition of the $n$-torus (the winding partition) and directly result from the network structure of the system. The concepts of winding numbers and winding cells are illustrated in Fig. 1.

In this article, we rigorously draw the link between the winding cells and the occurrence of a series of surprising behaviors of dissipative oscillator systems, that have escaped analysis up to this day. Specifically, we show that, in some cases, increased dissipation can lead to more robust and more stable systems. We argue that the winding partition provides a clear phase portrait for the analysis of such behaviors.

Motivated by these first observations, we proceed to the second contribution of this article. Namely, we provide an analytical, statistical, and computational understanding of the solutions of dissipative flow problems. In particular, we show that, exactly as in lossless networks, there is at most one solution of the dissipative flow problem in each winding cell. This at most uniqueness property is rigorously proven for a small amount of dissipation and verified numerically for a wide range thereof. For acyclic networks, we provide an algorithm

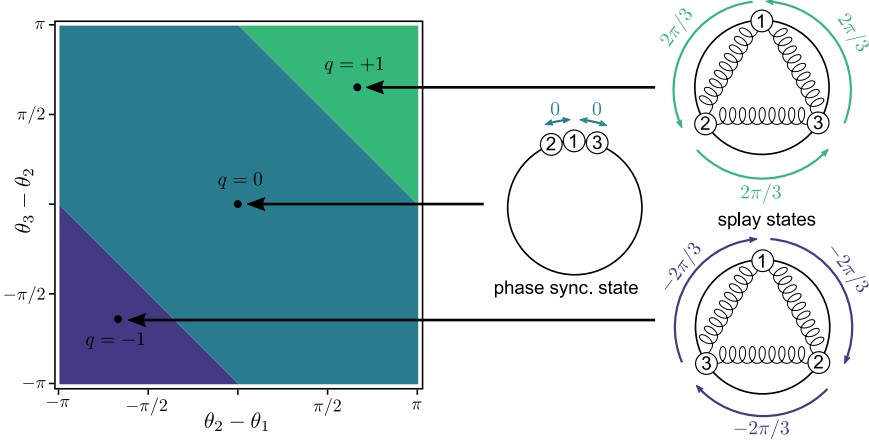

**Fig. 1 | Winding cell partition and equilibria for a three-node spring network.** Projected winding cell partition of the 3-torus (left) and representation of three equilibria of a spring network (right). Points on the 3-torus are projected on the two-dimensional space of angular differences $\theta_2 - \theta_1$ and $\theta_3 - \theta_2$. A winding number $q \in \mathbb{Z}$ is associated to each equilibrium, counting the number of times its angles wind counterclockwise around the origin. The three dots (left) are equilibria of the spring network, labeled with their winding numbers $q$. The phase synchronous state has all angles identical and, therefore $q = 0$. Equilibria with $q = \pm 1$ are the so-called splay states or twisted states. The colored areas in the left panel represent the set of points with the same winding number, i.e., the winding cells, forming a partition of the 3-torus.

computing the unique solution, if it exists. For general, cyclic graphs, we provide an iteration map that, under technical assumptions, converges to the unique solution in a given winding cell. An implementation of both algorithms is provided freely online[41].

We conclude by providing a compelling example of a realistic power grid where multiple solutions of the power flows are accurately captured by the distinct winding cells. Namely, we find two power flow solutions on the IEEE RTS-96 test system[42], belonging to two different winding cells.

Overall, in this article, we illustrate our findings with the Kuramoto–Sakaguchi model (see the Methods Section). This model is particularly appealing in our context: it is a natural extension of the (lossless) Kuramoto model; it is a first-order version of the lossy swing equations; and the amount of dissipation in the coupling can easily be tuned with a continuous parameter, namely the phase frustration $\phi \in \mathbb{R}$. Nevertheless, our analytical results are valid for a much broader class of coupling functions $f_{ij} + g_{ij}$, that will be of interest to the dynamical systems and network science communities.

**Remark.** In addition to the demonstrations provided, the framework proposed here is naturally suited to the analytical study of networked dynamical systems with directed interactions. For sake of conciseness and clarity, we limit our focus to dissipative interactions over undirected edges, but the framework covers naturally any type of directed interactions. We discuss these generalizations to a greater extent in the Discussion Section.

## Results

After a formal definition of the winding partition of the $n$-torus, we provide a careful description of a series of unexpected behaviors of dissipative oscillator networks. This section culminates with a presentation of our rigorous mathematical results. We conclude by giving an example of two solutions of the power flow equations, coexisting on the IEEE RTS-96 test system. A detailed formalism can be found in the Methods Section and proofs are deferred to the Supplementary Information.

### Algebraic graph theory on the torus

Our framework is inspired by ref. 20. The states of the system of equation (2) are points $\boldsymbol{\theta}$ in the $n$-torus $\mathbb{T}^n$, each component being a point $\theta_i$ of the circle $\mathbb{S}^1$. Comparing points on $\mathbb{S}^1$ requires to define angular differences, which is somewhat arbitrary. In this article, we use the counterclockwise difference

$$d_{cc}(\theta_1, \theta_2) = \mathrm{mod}(\theta_1 - \theta_2 + \pi, 2\pi) - \pi \in [-\pi, \pi). \quad (5)$$

Intuitively, the counterclockwise difference is a projection of the angular difference on the interval $[-\pi, \pi)$.

Given a cycle $\sigma = (i_1, \ldots, i_{|\sigma|}, i_1)$ in a graph $G_u$ (see Methods for details), one can calculate the winding number around cycle $\sigma$ associated with the state $\boldsymbol{\theta} \in \mathbb{T}^n$,

$$q_\sigma(\boldsymbol{\theta}) = (2\pi)^{-1} \sum_{j=1}^{|\sigma|} d_{cc}(\theta_{i_j}, \theta_{i_{j+1}}) \in \mathbb{Z}. \quad (6)$$

Three states with different winding numbers are illustrated in Fig. 1 for the 3-cycle. Intuitively, the winding number counts the number of times the angles in $\boldsymbol{\theta}$ wind around the origin when following the cycle $\sigma$. Then, given a cycle basis $\Sigma = \{\sigma_1, \ldots, \sigma_c\}$ of the graph, we naturally define the winding vector associated to a state $\boldsymbol{\theta} \in \mathbb{T}^n$,

$$\mathbf{q}_\Sigma(\boldsymbol{\theta}) = \left[ q_{\sigma_1}(\boldsymbol{\theta}), \ldots, q_{\sigma_c}(\boldsymbol{\theta}) \right]^\top \in \mathbb{Z}^c. \quad (7)$$

Nonzero winding numbers are typically associated to loop flows[20,43,44], i.e., a commodity flow of constant magnitude around a cycle of the network. Such loop flows occupy line capacity, but do not deliver commodity anywhere.

**Remark.** The winding number is a natural extension to complex networks of the quantification of vortex flows in regular lattices, that arise in statistical physics [e.g., superfluids[45] or superconductors[46]]. As far as we can tell, the notion of winding numbers in systems of coupled oscillators can be traced back to refs 47. (referee discussion) and 48.

For a graph with $c$ cycles, a winding vector $\mathbf{u} \in \mathbb{Z}^c$ can be uniquely associated with each state in $\mathbb{T}^n$. Therefore we can define the winding cell associated with winding vector $\mathbf{u}$,

$$\Omega(\mathbf{u}; \Sigma) = \left\{ \boldsymbol{\theta} \in \mathbb{T}^n : \mathbf{q}_\Sigma(\boldsymbol{\theta}) = \mathbf{u} \right\}. \quad (8)$$

The counterclockwise difference is bounded, and so are the winding numbers. There is then a finite number of winding cells for a given graph $G_u$, forming a finite partition of $\mathbb{T}^n$. See Fig. 2 for an illustration of winding cells in a cycle of $n = 3$ oscillators.

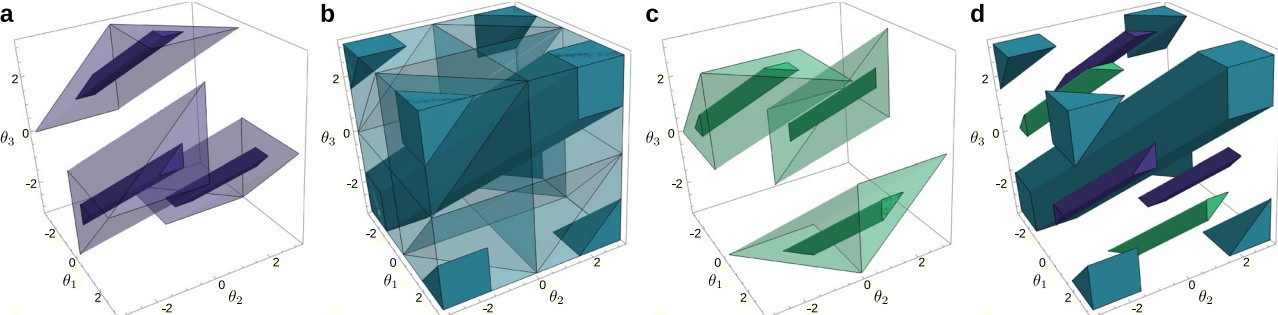

**Fig. 2 | Winding cells and cohesive sets in a three-node system.** Winding cells and their cohesive subsets, for a cycle of length $n = 3$. The three-dimensional plots show the unfolded 3-torus, where each dimension parametrizes one of the three angles and the winding cells become polytopes. The sides of the cube have then to be considered as identified (left-right, top-bottom, front-back). **a** The transparent volume is $\Omega(+1; \sigma)$, the winding cell of winding number $q = +1$, and the solid volume is the $3\pi/4$-cohesive set, i.e., the subset of $\Omega(+1; \sigma)$ where the counterclockwise differences do not exceed $3\pi/4$. **b** The transparent volume is $\Omega(0; \sigma)$, the winding cell of winding number $q = 0$, and the solid volume is the $\pi/2$-cohesive set. **c** The transparent volume is $\Omega(-1; \sigma)$, the winding cell of winding number $q = -1$, and the solid volume is the $3\pi/4$-cohesive set. **d** Union of the cohesive sets of the previous panels. Each color corresponds to a winding cell. A key result of this article is that there is at most one solution to equation (2) in a certain cohesive subset of each winding cell, i.e., in each solid volume.

## Anomalous behaviors of dissipative systems

Three unexpected behaviors of lossy oscillator networks are illustrated in Fig. 3 for the Kuramoto–Sakaguchi model. The first one is a direct extension of a phenomenon already noted for lossless systems[20,49]. The two other have not been reported to the best of our knowledge.

Anomaly 1, loop flows increase capacity: One would expect the presence of a loop flow (i.e., nonzero winding number) to reduce transmission capacity of the system, because lines are occupied by the aforementioned loop flow. In Fig. 3c and f, we see that the solutions with larger winding numbers tolerate larger commodity transfers. Even though such observations have been documented in the past for lossless systems[20,49], it remains somewhat counterintuitive.

Anomaly 2, dissipation increases capacity: An initial reasoning would suggest that increasing dissipation would reduce the robustness and reliability of a system. Indeed, if part of the transmitted commodity is lost on the way, then more of it needs to be injected and the system is operated closer to criticality. However, the relation between dissipation and robustness is not that simple, as we illustrate in Fig. 3c. Indeed, for a nonzero winding number, the ability of the system to synchronize can evolve non-monotonously with respect to the dissipation (see solution at $q = -1$). Such a phenomenon is quite unexpected and, to the best of our knowledge, has not been reported so far.

Anomaly 3, dissipation promotes multistability: Different solutions differ by a collection of loop flows[43], i.e., for some solutions, the lines are more loaded than for others. Similarly, as in the previous anomaly, one would expect that increased dissipation would prevent the occurrence of loop flows and, therefore of multiple solutions. However, according to Fig. 3f, a system with low dissipation ($\phi \in [0, 0.3]$) and low injection ($p \approx 0$) can have fewer solutions than more loaded and dissipative systems. Indeed, one would assume that lower loads and lower frustration leads to a larger margin of freedom in the system. Apparently, this is not necessarily the case and this can be attributed to the underlying network structure.

The anomalous behaviors identified above are typically related to the coexistence of different solutions. As we show in this article, there is a strong and direct link between different solutions and the winding partition of the $n$-torus.

## Problem setup and solution: synchronous states with dissipative couplings

We now formalize the problem of flow distribution in dissipative networks and present our main formal results. We provide a summary of the main notation symbols in the Methods Section (Table 1).

Let $G_u$ be the undirected graph describing the interactions in equation (2). Each edge $e = \{i, j\}$ of $G_u$ is endowed with two coupling functions,

$$h_{ij}(x) = f_{ij}(x) + g_{ij}(x), \quad h_{ji}(x) = f_{ji}(x) + g_{ji}(x), \tag{9}$$

one for each orientation. Without loss of generality, we incorporate the edge weight $a_{ij}$ in the coupling functions $f_{ij}$, $g_{ij}$, and $h_{ij}$. For each edge, we choose an arbitrary orientation, say $(i, j)$. We will refer to the coupling functions as $h_e = h_{ij}$ and $h_{\bar{e}} = h_{ji}$, with $\bar{e}$ denoting edge $e$ with reversed orientation.

In our framework, equation (2) can be written in a vectorial form as

$$\begin{aligned} M\ddot{\boldsymbol{\theta}} + D\dot{\boldsymbol{\theta}} &= \boldsymbol{\omega} - B_o \left[ \mathbf{f}(B_u^\top \boldsymbol{\theta}) + \mathbf{g}(B_u^\top \boldsymbol{\theta}) \right] \\ &= \boldsymbol{\omega} - B_o \mathbf{h}(B_u^\top \boldsymbol{\theta}), \end{aligned} \tag{10}$$

using the diagonal inertia and damping matrices $M$ and $D$, the incidence matrix $B_u$, and the out-incidence matrix $B_o$ induced by the chosen orientation for the incidence matrix [Methods Section, equations (21) and (25)]. The coupling function $\mathbf{h} : \mathbb{R}^m \to \mathbb{R}^{2m}$ relates a vector of angular differences over the $m$ undirected edges to the flows that are distinct for each edge orientation, hence $\mathbf{h}$ has $2m$ components,

$$[\mathbf{h}(\mathbf{y})]_e = h_e(y_e), \quad [\mathbf{h}(\mathbf{y})]_{e+m} = h_{\bar{e}}(-y_e). \tag{11}$$

The edge indices $e \in \{1, ..., m\}$ follow the orientation induced by the incidence matrix $B_u$. We refer to the discussion about the Kuramoto–Sakaguchi model in the Methods Section for an instructive example of the construction of equation (10).

From now on, it will be convenient to formulate the problem in terms of angular difference variables $\boldsymbol{\Delta} \in \mathbb{R}^m$, rather than in terms of angle variables $\boldsymbol{\theta} \in \mathbb{R}^n$. Constructing the vector of angular differences $\boldsymbol{\Delta}$ from a vector of angles $\boldsymbol{\theta}$ is straightforward, using the transpose of the incidence matrix, $\boldsymbol{\Delta} = B_u^\top \boldsymbol{\theta}$. The other direction, however is not that direct. Indeed, from a difference vector $\boldsymbol{\Delta}$, one can recover the associated angle vector $\boldsymbol{\theta}$ over a spanning tree of the graph. Now, the angular difference vector is consistent with the graph structure only if some cycle constraints are satisfied. Namely, over the remaining edges of the graph $e = \{i, j\}$, that are not in the spanning tree, the constraint is $\theta_i - \theta_j = \Delta_e + 2\pi k$, $k \in \mathbb{Z}$. The integer multiple of $2\pi$ does not matter because the angles are compact variables over $\mathbb{S}^1$. Mathematically speaking, these cycle constraints can be formalized using the cycle-edge incidence matrix $C_\Sigma$ associated with a cycle basis $\Sigma = (\sigma_1, ..., \sigma_c)$ [formally defined in the Methods Section, equation (23)]

$$C_\Sigma \boldsymbol{\Delta} = 2\pi \mathbf{u}, \tag{12}$$

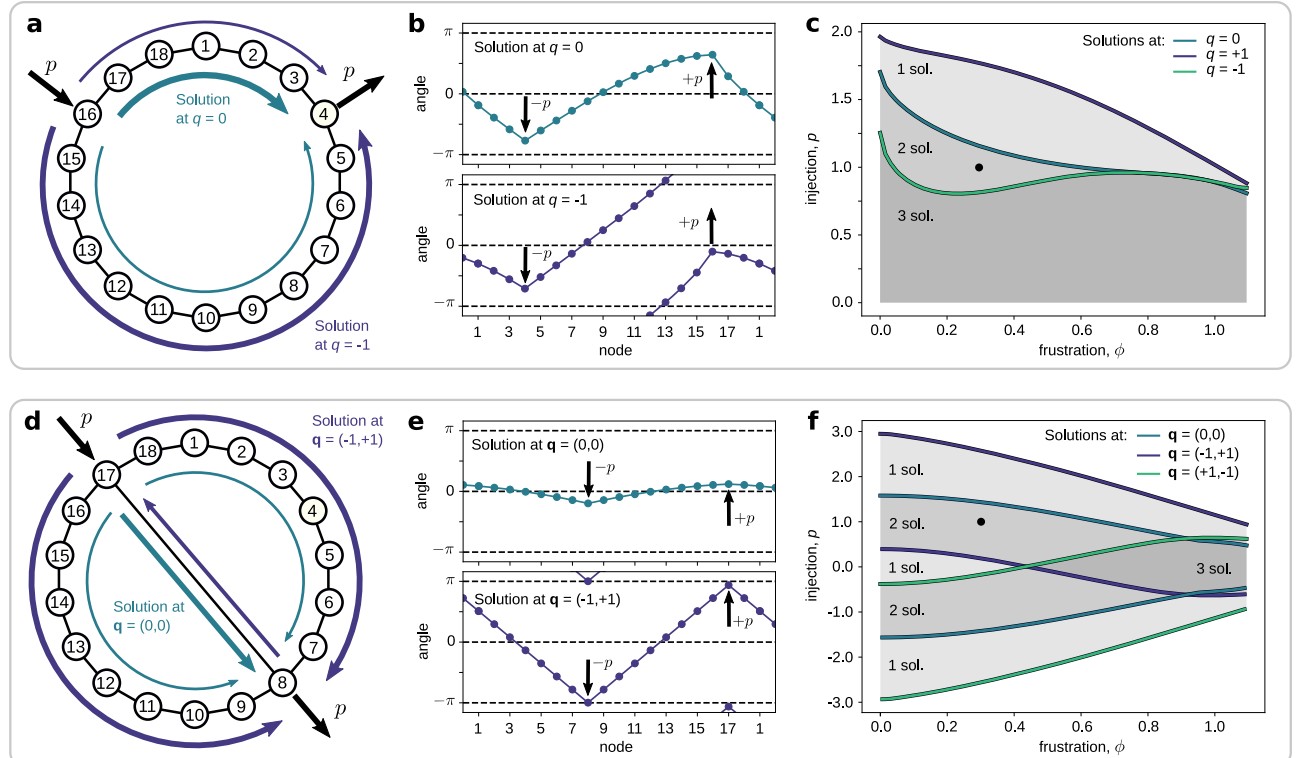

**Fig. 3 | Anomalous behaviors in the Kuramoto–Sakaguchi model.** Illustration of the anomalous behaviors identified for the Kuramoto–Sakaguchi model on a cycle network (**a**–**c**) and a two-cycle network (**d**–**f**), with unit coupling weights. **a** Qualitative distributions of flows over a cycle of $n = 18$ oscillators, with commodity injection $+p$ (resp. withdrawal $-p$) at node 16 (resp. 4). The arrows of two different colors visualize different flow solutions, with different winding numbers. **b** Angles corresponding to the two solutions of panel **a**. One clearly sees that, for the solution at $q = +1$, the angles wrap around the circle, but not in the solution at $q = 0$. **c** Boundaries (colored curves) of the existence regions for solutions at different winding numbers, in the parameter space of phase frustration $\phi$ and injection magnitude $p$. The solutions in panels **a** and **b** were obtained for $(\phi, p) = (0.3, 1.0)$. The darkness of each area in the parameter space represents the number of existing synchronous states. It is surprising that (i) the solution at $q = +1$, i.e., with a larger winding number, can carry a larger flow than the solution at $q = 0$, and (ii) for the solution at $q = -1$, the maximal tolerated commodity injection is not monotone in the frustration. **d**–**f** Same (**a**–**c**) respectively, for the two-cycle network in panel **d**. **f** Surprisingly, this network has fewer solutions for light load and frustration, $(\phi, p) \approx (0.0, 0.0)$, rather than for larger parameter values, e.g., $(\phi, p) = (0.3, 1.0)$.

for some winding vector $\mathbf{u} \in \mathbb{Z}^c$. In summary, an angular difference vector $\mathbf{\Delta}$ needs to satisfy equation (12) in order to be consistent with an angle vector $\boldsymbol{\theta}$.

From now on, we will search for stable synchronous states of equation (10), i.e., we require the Jacobian matrix of the system to have its spectrum in the left complex half-plane[50]. Following Gershgorin Circles Theorem[51], stability is guaranteed if $h'(\Delta_e), h'_{\bar{e}}(-\Delta_e) > 0$ for all $e$. Formally, we assume that, in a neighborhood of the origin, both $h_e$ and $h_{\bar{e}}$ are strictly increasing, and for each edge $e$ of $G_u$, we require $|\Delta_e| \leq \gamma_e$, so derivatives are positive. The vector of angular differences is then restricted to the hypercube

$$R(\boldsymbol{\gamma}) = \bigcap_{e \in E_u} [-\gamma_e, \gamma_e] \subset \mathbb{R}^m. \qquad (13)$$

The set of points $\boldsymbol{\theta} \in \mathbb{T}^n$ whose angular differences along the edges of $G_u$ are in $R(\boldsymbol{\gamma})$ is referred to as a $\boldsymbol{\gamma}$-cohesive set. The solid volumes in Fig. 2 show the intersections of the various winding cells of a 3-cycle and $R(\boldsymbol{\gamma})$ for different values of $\gamma_e$.

Gathering the above observations, we formulate the following problem, whose solutions are in one-to-one correspondence with synchronous states of equation (10).

**Problem statement.** (Dissipative Flow Network). Given a connected graph $G_u$ with $n$ nodes, $m$ edges, and cycle basis $\Sigma$, a vector of natural frequencies $\boldsymbol{\omega} \in \mathbb{R}^n$, and appropriate coupling functions $h_e, h_{\bar{e}}$,

associated to each edge $e$, find a solution $\mathbf{\Delta} \in R(\boldsymbol{\gamma})$ of

$$B_o \mathbf{h}(\mathbf{\Delta}) - \boldsymbol{\omega} = \varphi \mathbf{1}_n, \qquad (14a)$$

$$C_\Sigma \mathbf{\Delta} = 2\pi \mathbf{u}, \qquad (14b)$$

for some synchronous frequency $\varphi \in \mathbb{R}$ and winding vector $\mathbf{u} \in \mathbb{Z}^{m-n+1}$.

In contrast with previous works on lossless systems, the flow map $h_e$ is not odd, meaning that we do not impose the constraint $h_e(\theta_i - \theta_j) = -h_{\bar{e}}(\theta_j - \theta_i)$, hence our need of the out-incidence matrix $B_o$ in equation (14a). Note that, even though in our example of the Kuramoto–Sakaguchi model all coupling functions are identical, in full generality, we allow $h_e \neq h_{\bar{e}}$.

In the Methods Section, we provide a series of rigorous results, proving the following claim.

**Claim.** There is at most one solution to the dissipative flow network problem in each winding cell of the $n$-torus.

More precisely, we distinguish the cases of acyclic graphs and of general graphs with cycles. For acyclic graphs, the winding partition is trivial and the whole $n$-torus is a single winding cell. By proving that there is at most one solution to the Dissipative Flow Network problem for such graphs, the claim is proven (see Theorem 3).

In the case of a cyclic graph, we design an iterative map that we prove to be contracting within winding cells, under reasonable

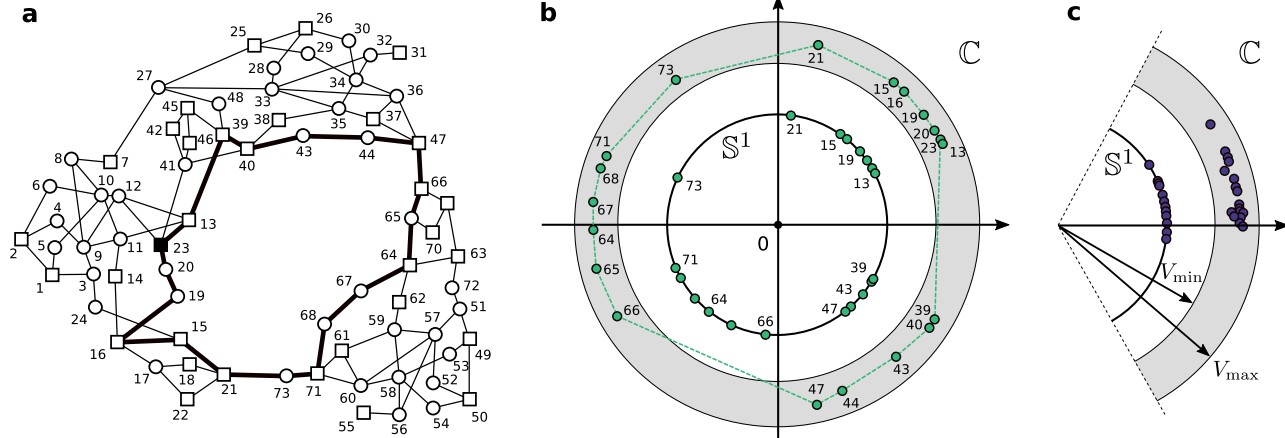

**Fig. 4 | Multistability in the full power flow equations for the IEEE RTS-96 test case.** Comparison of the power flow solutions and Kuramoto–Sakaguchi synchronous states on the IEEE RTS-96 test case[42]. **a** Geographic representation of the system. Circles are loads and squares are generators. The network is composed of $n = 73$ nodes, $m = 108$ edges, and therefore $c = 36$ independent cycles. The long cycle with thick edges is of particular interest, because its length promotes the existence of loop flows while keeping angular differences small [see refs. 43, 44 for an extended discussion]. **b, c** Combined representations of: (outer annulus) the complex voltages for solutions to the full power flow equations for an adapted version of the IEEE RTS-96 test case; (inner circle) the phase angles of synchronous states of the Kuramoto–Sakaguchi model on the same system. For the sake of readability, only the values of the nodes around the long cycle of panel **a** are represented. The outer annulus represents the tolerated margin of variation for the voltage amplitudes in the power flow equations. The power flow solution in panel **b** has a nonzero winding number ($q = +1$) and there is a reasonable correspondence (ordering, clustering) between its voltage phases and the angles of the Kuramoto–Sakaguchi synchronous state. Similarly, both the power flow solution and the Kuramoto–Sakaguchi synchronous states in panel **c** have zero winding number, with all angles in a relatively short arc.

conditions (see Corollary 6). Solutions to the Dissipative Flow Network problem are fixed points of this iteration map, and therefore, there is at most one solution in each winding cell.

The formal results are formulated in the Methods Section and proof are deferred to the Supplementary Information.

### Multistability in power grids

The last decades have seen a large-scale effort of the complex systems community to provide an analytical description of the power flow equations and of their solutions (see the Methods Section). In 1972 already, Korsak[47] showed that, mathematically speaking, the power flow equations tolerate multiple solutions on cyclic networks. Since then, there has been a plethora of evidence, both analytical and numerical, that the power flow equations allow the coexistence of different solutions[13,52–55]. Even some "real-world" events advocate in this direction[56,57]. However, a large proportion of the work mentioned above relies on the lossless line assumption, namely, neglecting dissipation, voltage amplitude dynamics, and reactive power flows. Recently, there has been a common effort in trying to pursue a more realistic mathematical analysis of power grids, by incorporating reactive power flows[18,19], voltage amplitude dynamics[58,59], and dissipation[27,28]. In particular, the recent linear stability analysis of the extended swing equations proposed in ref. 60. show that the conditions for the local stability of a synchronous state in lossy systems are very similar to those for lossless systems. Despite all this work, there is still neither a clear extension of the winding partition to the full active-reactive power flows, nor a global phase portrait for lossy oscillator systems, even though there are some notable related preliminary works[61,62]. Our results are an advance in the aforementioned collective effort.

To put our results in perspective with the resolution of the power flow equations, we solved both the Dissipative Flow Network Problem and the power flow equations on an adapted version of the IEEE RTS-96 test case[41,42]. In Fig. 4, we compare synchronous states of the Kuramoto–Sakaguchi model (panels b and c, inner circle), with the corresponding solutions of the full power flow equations (panels b and c, outer annulus). We elaborate on the resolution of the full power flow equations in the Methods Section. First of all, one clearly sees that the main qualitative features (e.g., winding number, cohesiveness, clustering) of the power flow solutions are captured by the corresponding synchronous states of the Kuramoto–Sakaguchi model. Furthermore, it is remarkable that two solutions to the full power flow equations coexist, satisfying all voltage amplitude constraints as well as voltage angle stability. This example shows that loop flows and winding partition are fundamental features of power flow solutions. Our work is a contribution to the joint and long-lasting effort in the quest for an accurate mathematical analysis of power grids, which is a landmark in the area of power grid analysis.

## Discussion

Theorem 3 and Corollary 6 (see Methods Section) rigorously prove that, in each winding cell of the $n$-torus, there is at most a unique synchronous solution for dissipative networks of oscillators. In acyclic networks, the whole $n$-torus is trivially the unique winding cell, and there is, therefore, at most one solution to the Dissipative Flow Network problem (Theorem 3), independently of the amount of dissipation. For systems over more general, cyclic graphs, the winding partition provides a natural decomposition of the $n$-torus in subsets containing at most one solution. These results are a straight generalization of ref. 20 to dissipative systems.

Even though the relation established in Corollary 6 is formally valid for relatively small amounts of dissipation, numerical experiments did not lead to any counterexample. Indeed, we empirically observed for a large range of network structures, frustration parameters, and initial conditions, that the iteration map defined in Theorem 5 [Methods Section, equation (37)] can always be made contracting by taking a sufficiently small value of $\epsilon > 0$. We have, therefore, strong numerical evidence that the aforementioned iteration map can be made contracting at any point of the $\gamma$-cohesive set, with $\gamma_e$ taken such that each coupling function $h_e$ is strictly increasing on $[-\gamma_e, \gamma_e]$. We conjecture that Corollary 6 is actually very conservative in general and that the at most uniqueness property therein is valid for a much broader range of dissipation-to-coupling ratio. Furthermore, the comparison between coupling and dissipation in equation (45) clearly pinpoints how dissipation works against synchronization.

When considering lossless couplings, one realizes that the inequality in (45) is always satisfied, recovering the results of ref. 20.

Both the proofs of Theorem 3 and Corollary 6 are algorithmic by nature. Namely, the proof of Theorem 3 considers recursively the flows on the edges of the acyclic graph, and Corollary 6 relies on an iteration map [$S_e$, equation (37)]. It is therefore straightforward to actually implement the proofs as routines, which we provide online[41].

The anomalies illustrated in Fig. 3 emphasize that the introduction of dissipation in the coupling between oscillators has a nontrivial and surprising impact on the dynamics. The fact that both loop flows and dissipation can increase the transmission capacity of a system (Anomalies 1 and 2) is arguably counterintuitive. We remark that both Anomalies 1 and 2 occur for solutions with nonzero winding numbers ($q = +1$ and $q = -1$, respectively). It is also quite unexpected that more loaded and dissipative systems can possess more flow network solutions for a given network structure (Anomaly 3). Again, this last anomaly involves solutions in different winding cells. All anomalies identified in Fig. 3 are strongly linked to solutions with nontrivial winding numbers. A general and thorough description of the different operating states of dissipative networks of oscillators is then required to tackle these systems through the prism of the winding partition. On top of that, through our realistic example on the IEEE RTS-96 test system, we show that the winding partition will be relevant in the analysis of multiple solutions to the full power flow equations.

We trust that the notion of winding partition has the potential to contribute elucidating many open problems in the fascinating phenomenon of synchronization in complex networks. We reiterate that even though we restricted our discussion to bidirected interactions for the sake of clarity, the whole framework developed in this article naturally applies to any system with directed interactions. Namely, our formalism is the first step towards a unified analysis of synchronization in any network of coupled oscillators, no matter the nature of the interactions.

## Methods

We first provide the necessary grounds of directed and undirected graph theory, as well as a link between them. We point to ref. 16. for an extended discussion about graph and digraph theory. We then discuss the Kuramoto–Sakaguchi model and its link with the power flow equations. We conclude this section with the formulation of our theoretical results.

### Directed graphs

A directed graph (or digraph) $G_d$ is the pair $(V, E_d)$ composed of a set of vertices (or nodes) $V = \{1, ..., n\}$ and a set of directed edges $E_d \subset V \times V$, which are ordered pairs of vertices. For an edge $e = (i, j) \in E_d$, $i$ is the source of $e$, denoted $s_e$, and $j$ is its target, denoted $t_e$, i.e., $e = (s_e, t_e)$. We denote the edge with opposite direction as $\bar{e} = (t_e, s_e)$. The existence of edges is encoded in the graph's adjacency matrix

$$(A_d)_{ij} = \begin{cases} 1, & \text{if } (i,j) \in E_d, \\ 0, & \text{otherwise.} \end{cases} \tag{15}$$

The out-degrees (resp. in-degrees) matrix is obtained as $D_o = \text{diag}(A_d \mathbf{1})$ (resp. $D_i = \text{diag}(A_d^\top \mathbf{1})$). We define the Laplacian matrix of $G_d$ as $L_d = D_o - A_d$. For a digraph with $n$ vertices and $m$ directed edges, we define the $n \times m$ out-incidence and in-incidence matrices

$$(B_o)_{ie} = \begin{cases} 1, & \text{if } e = (i,j) \text{ for some } j, \\ 0, & \text{otherwise,} \end{cases} \tag{16}$$

$$(B_i)_{ie} = \begin{cases} 1, & \text{if } e = (j,i) \text{ for some } j, \\ 0, & \text{otherwise,} \end{cases} \tag{17}$$

**Table 1 | List of symbols**

| Symbol | Name/Description |
|---|---|
| $V$ | Set of vertices. |
| $G_u, E_u$ | Undirected graph, undirected edge set. |
| $B_u, L_u$ | Incidence and Laplacian matrices of an undirected graph [equation (21)]. |
| $C_\Sigma$ | Cycle-edge incidence matrix of the set of cycles $\Sigma$ [equation (23)]. |
| $G_d, E_d$ | Directed graph, set of directed edges. |
| $G_b, E_b$ | Bidirected counterpart of the undirected graph $G_u$ and its set of directed edges. |
| $s_e, t_e$ | Source and target of edge $e$. |
| $\bar{e}$ | Edge $e$ with opposite direction. |
| $A_d, B_d, L_d$ | Adjacency, incidence, and Laplacian matrices of a digraph [equations (15), (18)]. |
| $B_b$ | Incidence matrix of a bidirected graph. |
| $B_o, B_i$ | Out- and in-incidence matrices of a digraph. |
| $\boldsymbol{\theta} = (\theta_1 \ldots \theta_n)^\top$ | Vector of phase angles. |
| $\boldsymbol{\omega} = (\omega_1 \ldots \omega_n)^\top$ | Vector of natural frequencies. |
| $a_{ij}, \phi_{ij}$ | Coupling strength and phase frustration between nodes $i$ and $j$. |
| $\gamma_e$ | Bound on the angular difference over the edge $e$. |
| $h_e, h_{\bar{e}}$ | Coupling functions over the edge $e$. |
| $f_e, g_e$ | Odd and even parts of the coupling over edge $e$ [equation (38), (39)]. |
| $R(\boldsymbol{\gamma})$ | Domain of bounded angular differences [equation (13)]. |
| $\mathbf{q}_\Sigma$ | Winding map for the cycles in $\Sigma$ [equation (7)]. |
| $\Omega(\mathbf{u}; \Sigma)$ | Winding cell with winding vector $\mathbf{u}$ in the graph $G_u$ [equation (8)]. |
| $S_e$ | Flow mismatch iteration [equation (37)]. |

which form the standard incidence matrix

$$B_d = B_o - B_i. \tag{18}$$

We notice the following relations, the fourth being unknown as far as we can tell.

**Proposition 1.** The adjacency matrix $A_d$, the out- and in-degree matrices $D_o$ and $D_i$, and the Laplacian matrix $L_d$ of a directed graph can be written in terms of its out- and in-incidence matrices $B_o$ and $B_i$:

$$\begin{aligned} D_o &= B_o B_o^\top, & D_i &= B_i B_i^\top, \\ A_d &= B_o B_i^\top, & L_d &= B_o B^\top. \end{aligned} \tag{19}$$

**Proof.** The proofs for the adjacency matrix $A_d$ and for the degree matrices $D_o$ and $D_i$ can be found in ref. 63. (Lemmas 3.1 and 4.1). The proof for the Laplacian matrix directly follows,

$$\begin{aligned} L_d &= D_o - A_d = B_o B_o^\top - B_o B_i^\top \\ &= B_o (B_o - B_i)^\top = B_o B^\top. \end{aligned} \tag{20}$$

**Remark.** The same proof is straightforwardly adapted to weighted directed graphs.

### Undirected graphs

An (undirected) graph $G_u$ is a pair $(V, E_u)$ composed of a set of $n$ vertices (or nodes) $V = \{1, ..., n\}$ and a set of $m$ edges, which are unordered pairs

of vertices, $E_u \subset \{\{i,j\} : i,j \in V\}$. A cycle of $G_u$ is an ordered sequence of vertices $\sigma = (i_0, i_1, ..., i_\ell = i_0)$, such that $\{i_j, i_{j+1}\} \in E_u$ and $i_j \neq i_k$ for any $j, k \in \{1, ..., \ell\}$.

Let us now choose an arbitrary orientation [$(i, j)$ or $(j, i)$] for each undirected edge $\{i,j\} \in E_u$. We can then define the incidence matrix of $G_u$,

$$(B_u)_{ie} = \begin{cases} 1, & \text{if } e = (i,j) \text{ for some } j, \\ -1, & \text{if } e = (j,i) \text{ for some } j, \\ 0, & \text{otherwise.} \end{cases} \quad (21)$$

The Laplacian matrix of $G_u$ can be computed as $L_u = B_u B_u^\top$. Note that the incidence matrix is not unique and depends on the choice of edge orientations, whereas the Laplacian does not. Given a cycle $\sigma$, we define the cycle vector $\mathbf{v}_\sigma \in \{-1, 0, +1\}^m$, indexed by edges, as

$$(\mathbf{v}_\sigma)_e = \begin{cases} +1, & \text{if } e = (i_{k-1}, i_k) \text{ for some } k, \\ -1, & \text{if } e = (i_k, i_{k-1}) \text{ for some } k, \\ 0, & \text{otherwise.} \end{cases} \quad (22)$$

The cycle space of $G_u$ is the span of the cycle vectors of all cycles of $G_u$, which is equivalently defined as the kernel of the incidence matrix $B_u$. A set of cycles $\Sigma = \{\sigma_1, ..., \sigma_c\}$ is a cycle basis of $G_u$ if and only if the set of corresponding cycle vectors forms a basis of the cycle space.

Finally, given a cycle basis $\Sigma$ of the graph $G_u$, we define the cycle-edge incidence matrix,

$$C_\Sigma = \left( \mathbf{v}_{\sigma_1}, \cdots, \mathbf{v}_{\sigma_c} \right)^\top \in \mathbb{R}^{c \times m}. \quad (23)$$

### Bidirected graphs

Dissipative couplings intrinsically require to distinguish the two orientations of each edge. Given an undirected graph $G_u = (V, E_u)$, its bidirected counterpart is the directed graph $G_b = (V, E_b)$ with the same vertex set $V$ and where each undirected edge $\{i,j\} \in E_u$ is doubled in the set of directed edges $(i, j), (j, i) \in E_b$. A bidirected graph is a directed graph induced by an undirected graph.

If the undirected graph $G_u$ has incidence matrix $B_u \in \mathbb{R}^{n \times m}$ [equation (21)], then, with appropriate indexing of the directed edges, the incidence matrix of $G_b$ can be written as $B_b = (B_u, -B_u) \in \mathbb{R}^{n \times 2m}$. Interestingly, we note that the Laplacian matrices of $G_u$ and $G_b$ are the same, namely (see Prop. 1),

$$L_u = B_u B_u^\top = L_b = B_o B_b^\top, \quad (24)$$

where the out-incidence matrix $B_o = [B_b]_+$ is the positive part of $B_b$. Notice that here,

$$B_o = \left( [B_u]_+, [B_u]_- \right). \quad (25)$$

### The Kuramoto–Sakaguchi model

We illustrate the results of this article with the generalized Kuramoto–Sakaguchi model [8,25,26],

$$\dot{\theta}_i = \omega_i - \sum_{j=1}^n a_{ij} \left[ \sin(\theta_i - \theta_j - \phi_{ij}) + \sin(\phi_{ij}) \right], \quad (26)$$

for $i \in \{1, ..., n\}$, where $\theta_i \in \mathbb{S}^1$ and $\omega_i \in \mathbb{R}$ are respectively the phase angle and the natural frequency of the $i$-th oscillator, $\phi_{ij} \in (-\pi/2, \pi/2)$ is the phase frustration between oscillators, and in this case, $a_{ij} \in \mathbb{R}_{\geq 0}$ is the coupling strength between oscillators $i$ and $j$. The Kuramoto–Sakaguchi model directly translates to the framework of equation (2), with $m_i = 0$, $d_i \equiv 1$, $f_{ij}(x) = a_{ij} \cos(\phi_{ij}) \sin(x)$, and

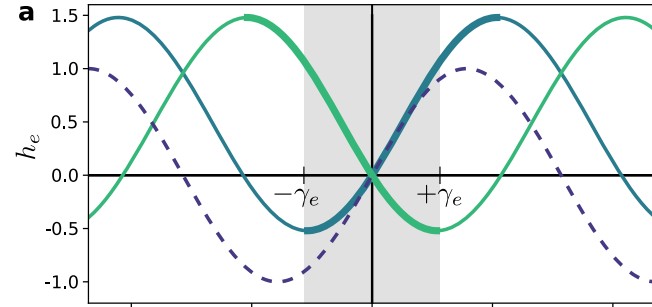

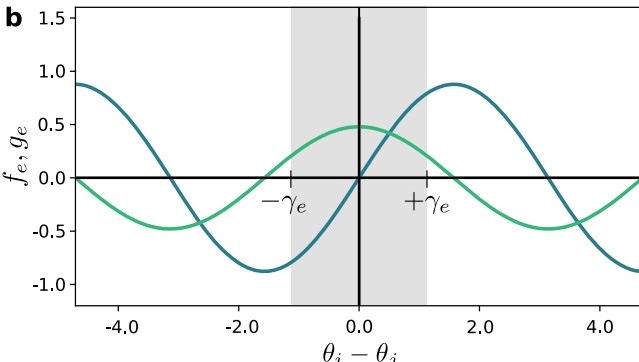

**Fig. 5 | Coupling functions for the Kuramoto–Sakaguchi model. a** Comparison between coupling functions for Kuramoto [dashed dark blue, $h_e(x) = \sin(x)$] and Kuramoto–Sakaguchi [plain cyan, $h_e(x) = \sin(x - \phi) + \sin(\phi)$], with $\phi = 0.5$. The light green curve illustrates the coupling on the same edge, but with opposite orientation [$h_e(-x) = \sin(-x - \phi) + \sin(\phi)$]. The thick parts (cyan and green) emphasize the region where the curve is increasing (resp. decreasing). The shaded gray area shows the interval where the coupling in both orientations is strictly monotone. **b** Odd (cyan) and even (green) parts of the Kuramoto–Sakaguchi coupling function, as defined in equation (44).

$g_{ij}(x) = a_{ij} \sin(\phi_{ij})[1 - \cos(x)]$. It is a natural extension of the original Kuramoto model, which is recovered for $\phi_{ij} = 0$. The coupling function of the Kuramoto–Sakaguchi model is illustrated in Fig. 5a.

**Remark.** While in its original formulation, the Kuramoto–Sakaguchi model assumes homogeneous, all-to-all couplings, here we take the couplings to be given by an underlying network structure. For the sake of simplicity, in our examples, we consider $a_{ij} = a_{ji}$ and $\phi_{ij} = \phi_{ji}$ for all connected nodes $i$ and $j$. Nevertheless, we keep in mind that these assumptions are not necessary for our results and that our framework is appropriate for much more general cases.

In order to illustrate some fundamental complications that arise in the Kuramoto–Sakaguchi model, compared to the original Kuramoto model, we detail two simple examples below.

**Example.** (2-node system, vectorial form). Consider a system of two coupled Kuramoto–Sakaguchi oscillators with unit coupling and identical frustration, whose dynamics is given by

$$\begin{aligned} \dot{\theta}_1 &= \omega_1 - \left[ \sin(\theta_1 - \theta_2 - \phi) + \sin(\phi) \right], \\ \dot{\theta}_2 &= \omega_2 - \left[ \sin(\theta_2 - \theta_1 - \phi) + \sin(\phi) \right], \end{aligned} \quad (27)$$

with $\phi \in (-\pi/2, \pi/2)$.

In the Kuramoto model ($\phi = 0$), it is standard to write the dynamics in vectorial form as

$$\dot{\boldsymbol{\theta}} = \boldsymbol{\omega} - B_u \mathcal{A} \sin\left( B_u^\top \boldsymbol{\theta} \right), \quad (28)$$

where $B_u \in \mathbb{R}^{n \times m}$ is the incidence matrix of the (undirected) coupling graph of the system, and $\mathcal{A} \in \mathbb{R}^m$ is the diagonal matrix of the edge

weights $a_{ij}$. In the Kuramoto–Sakaguchi model, this vectorial form is not that simple. Direct computation shows that naively writing

$$\dot{\boldsymbol{\theta}} = \boldsymbol{\omega} - B_u \mathcal{A}\left[\sin(B_u^\top \boldsymbol{\theta} - \phi \mathbf{1}_m) + \sin(\phi)\mathbf{1}_m\right], \qquad (29)$$

does not yields the desired equations (27).

In order to write equation (26) in vectorial form, we need to distinguish the two orientations of each edge and consider $G_b$, the bidirected counterpart of $G_u$. We need to introduce the out-incidence matrix $B_o$ and the incidence matrix $B_b$ of the bidirected coupling graph, as well as $\mathcal{A}_b \in \mathbb{R}^{2m}$, defined to be a block diagonal matrix whose two diagonal blocks are equal to $\mathcal{A}$. The proof of the following proposition follows from direct computation.

**Proposition 2.** The Kuramoto–Sakaguchi model in equation (26) is written in vectorial form as

$$\dot{\boldsymbol{\theta}} = \boldsymbol{\omega} - B_o \mathcal{A}_b\left[\sin(B_b^\top \boldsymbol{\theta} - \phi \mathbf{1}_{2m}) + \sin(\phi)\mathbf{1}_{2m}\right]. \qquad (30)$$

**Example.** (6-node cycle, sync. frequency). Let us consider six Kuramoto–Sakaguchi oscillators coupled in a cycle with identical, vanishing natural frequency, i.e.,

$$\dot{\theta}_i = -\sin(\theta_i - \theta_{i-1} - \phi) - \sin(\theta_i - \theta_{i+1} - \phi) + 2\sin(\phi), \qquad (31)$$

for $i \in \{1, \dots, 6\}$, where we used periodic indexing. One straightforwardly verifies that $\boldsymbol{\theta}_0 = (0, \dots, 0)^\top$ is an equilibrium of equation (31) (and then a synchronous state).

One can also verify that the splay state $\boldsymbol{\theta}_1 = (0, -\pi/3, -2\pi/3, \pi, 2\pi/3, \pi/3)^\top$ is also a synchronous state. Indeed, in this case, equation (31) gives

$$
\begin{aligned}
\dot{\theta}_i &= -\sin(-\pi/3 - \phi) - \sin(\pi/3 - \phi) - 2\sin(\phi) \\
&= 2\cos(\pi/3)\sin(\phi) - 2\sin(\phi) = -\sin(\phi),
\end{aligned} \qquad (32)
$$

independently of $i \in \{1, \dots, 6\}$. The state $\boldsymbol{\theta}_1$ is then synchronous, but it is an equilibrium only for the Kuramoto model ($\phi = 0$).

There are at least two main messages that can be taken from these examples. First, by extending our framework to directed graphs, we are able to write the Kuramoto–Sakaguchi model in vectorial from, in equations (30). Note that a similar vectorial formulation of the Kuramoto–Sakaguchi model has recently been proposed in ref. 64, which, while more general than equation (30), does not provide as much insight in the underlying network structure.

Second, unlike the Kuramoto model, the average frequency of the system is not preserved along arbitrary trajectories. Also, if multiple synchronous states exist, then they have, in general, different synchronous frequencies. These claims are backed up by showing that the average frequency of the system depends on angular differences,

$$\sum_i \dot{\theta}_i = \sum_i \omega_i - \sum_{i,j} a_{ij} \sin(\phi)\left[1 - 2\cos(\theta_i - \theta_j)\right], \qquad (33)$$

which is time-varying over the trajectories of the system and not identical for different synchronous states.

On top of that, we reiterate that, contrary to the Kuramoto model, the Kuramoto–Sakaguchi model is not the gradient of any function (even locally). Therefore, the energy landscape approaches, valid for $\phi = 0$[13,24], are not directly applicable when $\phi \neq 0$.

## The power flow equations

Under the assumption that voltage amplitudes are fixed, synchronous states of the Kuramoto–Sagauchi model are in direct correspondence with the solutions of the active power flow equations[13,65]. The power flow equations relate the the balance of active ($P_i$) and

reactive powers ($Q_i$) to the voltage amplitude ($V_i$) and phase ($\theta_i$) at each node $i \in \{1, \dots, n\}$,

$$P_i = \sum_{j=1}^n V_i V_j [\mathcal{B}_{ij} \sin(\theta_i - \theta_j) + \mathcal{G}_{ij}\cos(\theta_i - \theta_j)], \qquad (34)$$

$$Q_i = \sum_{j=1}^n V_i V_j [\mathcal{G}_{ij} \sin(\theta_i - \theta_j) - \mathcal{B}_{ij}\cos(\theta_i - \theta_j)], \qquad (35)$$

with $\mathcal{G}_{ij}$ and $\mathcal{B}_{ij}$ being lines conductance and susceptance, respectively. Defining

$$a_{ij} = V_i V_j \sqrt{\mathcal{B}_{ij}^2 + \mathcal{G}_{ij}^2}, \quad \phi_{ij} = \arctan(-\mathcal{G}_{ij}/\mathcal{B}_{ij}), \qquad (36)$$

one verifies that solutions of equation (34) are steady states of equation (26).

Equations (34) and (35) are usually solved by iterative methods. In Fig. 4b, c, outer annulus, we used a Newton–Raphson scheme[66] with different, carefully chosen initial conditions to solve the full power flow equations on our version of the IEEE RTS-96 test case[42]. The squares are PV buses, the circles are PQ buses, and the slack bus is node 23. The synchronous states of the Kuramoto–Sakaguchi models were computed by the flow mismatch iteration $S_\epsilon$ [equation (37)], with $\epsilon = 0.01$. All data are available online [41].

## The dissipative flow network problem on acyclic graphs

In the case where $G_u$ is acyclic, we show that there is at most a unique solution to the dissipative flow network problem. Here there are obviously no cycle constraints and thus equation (14b) is trivially satisfied.

**Theorem 3.** Consider the dissipative flow network problem on a connected acyclic undirected graph $G_u$. Then there is at most one $\boldsymbol{\Delta} \in R(\boldsymbol{\gamma})$ that satisfies equation (14a).

The proof of Theorem 3 proceeds recursively and we provide it in the Supplementary Information. An implementation of an algorithm deciding the existence of the unique solution is provided online [41].

**Remark.** Theorem 3 is the dissipative version of Theorem 2.2 in ref. 20. The spirit of Theorem 3 is somewhat similar to ref. 27, even though therein, the authors restrict their investigation to the Kuramoto–Sakaguchi model and cannot extend their approach to more general couplings.

## The dissipative flow network problem on general graphs

The presence of cycles in the network can induce the existence of multiple solutions to the dissipative flow network problem [see Fig. 3 or ref. 27]. We rigorously show here that winding vectors characterize these solutions for sufficiently moderate dissipation.

To do so, we define the flow mismatch iteration $S_\epsilon$ over the space of angular differences $\mathbb{R}^m$, whose fixed points are exactly the solutions of equation (14a). Namely, let

$$
\begin{aligned}
S_\epsilon : \mathbb{R}^m &\to \mathbb{R}^m \\
\boldsymbol{\Delta} &\mapsto \boldsymbol{\Delta} - \epsilon B_u^\top L_u^\dagger (B_o \mathbf{h}(\boldsymbol{\Delta}) - \boldsymbol{\omega}),
\end{aligned} \qquad (37)
$$

where $\epsilon > 0$ is a small step size and $L_u^\dagger$ is the pseudoinverse of the graph Laplacian matrix. The flow mismatch iteration $S_\epsilon$ updates the vector of angular differences according to the mismatch between the input/output of commodities $\boldsymbol{\omega}$ and the distribution of flows that corresponds to the current angular differences. It has two major properties:

(I)  the vector $\boldsymbol{\Delta}^* \in R(\boldsymbol{\gamma})$ is a fixed point of $S_\epsilon$ if and only if it is a solution of equation (14a);

(II) the map $S_\epsilon$ leaves each winding cell invariant, because $C_\Sigma B_u^\top = \boldsymbol{0}$. It means that fixing the winding vector of the initial

conditions imposes the winding vector of the fixed point of $S_\epsilon$, if ever it exists.

One of the main lessons from ref. [20] is that different solutions to the flow network problem on the $n$-torus are better understood when put in the context of their winding cell. Accordingly, and thanks to property II above, we split the dissipative flow network problem in each winding cell of the $n$-torus induced by the network structure. Fixing a winding vector $\mathbf{u} \in \mathbb{Z}^{m-n+1}$, we are guaranteed that if the initial conditions $\boldsymbol{\Delta}_0$ satisfy equation (14b), then each following iteration $\boldsymbol{\Delta}_{k+1} = S_\epsilon(\boldsymbol{\Delta}_k)$ will satisfy it as well.

We summarize the above observations in the following theorem, whose proof is a direct consequence of the compactness of $R(\boldsymbol{\gamma})$.

**Theorem 4.** If the flow mismatch iteration $S_\epsilon$ is contracting, then there is at most one synchronous state of equation (2) in each winding cell.

In what follows, we provide sufficient conditions for the contractivity of $S_\epsilon$. We rely on the decomposition of the coupling functions as $h_e(x) = f_e(x) + g_e(x)$, which implies,

$$f_e(x) = [h_e(x) - h_{\bar{e}}(-x)]/2, \tag{38}$$

$$g_e(x) = [h_e(x) + h_{\bar{e}}(-x)]/2. \tag{39}$$

One readily verifies the identities

$$f'_e(x) = f'_{\bar{e}}(-x), \qquad g'_e(x) = -g'_{\bar{e}}(-x). \tag{40}$$

Remind that $g_e$ quantifies to what extent the coupling is dissipative. In the particular case where the coupling is lossless, then $g_e = 0$.

Equipped with this decomposition of the couplings, we define two state-dependent matrices, for $\mathbf{x} \in R(\boldsymbol{\gamma})$:

(a) the odd weighted Laplacian matrix, which is the Laplacian matrix of $G_u$ weighed by the derivatives of the odd parts

$$L_f(\mathbf{x}) = B_u \cdot \mathrm{diag}[f'_e(x_e)] \cdot B_u^\top. \tag{41}$$

We emphasize that the choice of orientation for each edge $e$ does not matter in the definition of $L_f$. Also, the graph $G_u$ being connected and the above weights being positive, it is the standard result of algebraic graph theory that $\lambda_2$, the smallest nonzero eigenvalue of $L_f$ (a.k.a., the algebraic connectivity), is positive;

(b) the even weighted degree matrix, which is the diagonal matrix weighted by the absolute derivatives of the even parts,

$$\left[D_g(\mathbf{x})\right]_{ii} = \sum_{e \in E_i} |g'_e(x_e)|, \tag{42}$$

where $E_i$ is the set of (undirected) edges incident to node $i$. The diagonal terms of $D_g$ quantify the dissipativity of the couplings. In particular, for lossless couplings, $D_g = 0$.

**Example.** In the case of the Kuramoto–Sakaguchi model, the coupling functions are

$$h_e(x) = h_{\bar{e}}(x) = a_e[\sin(x - \phi) + \sin(\phi)], \tag{43}$$

and trigonometric identities yield

$$\begin{aligned} f_e(x) &= a_e \cos(\phi) \sin(x), \\ g_e(x) &= a_e \sin(\phi)[1 - \cos(x)], \end{aligned} \tag{44}$$

which we illustrate in Fig. [5]b. We clearly see here the relation between $g_e$ and the dissipativity or frustration of the coupling. When $\phi = 0$, we

recover the original Kuramoto model, where the coupling is lossless, and $g_e = 0$.

We are now ready to formulate the main theorem of this work. It clearly separates the impact of network connectivity, that promote the contractivity of $S_\epsilon$, and of the dissipation, that works against the contractivity of $S_\epsilon$. We defer the proof to the Supplementary Information.

**Theorem 5.** Given a dissipative flow network problem, define the odd weighted Laplacian $L_f$ and the even weighted degree matrix $D_g$. If, for all $i \in \{1, \ldots, n\}$,

$$\sup_{\mathbf{x} \in R(\boldsymbol{\gamma})} \left(D_g\right)_{ii} < \inf_{\mathbf{x} \in R(\boldsymbol{\gamma})} \lambda_2(L_f), \tag{45}$$

then there exists a sufficiently small step size $\epsilon > 0$ such that the flow mismatch iteration $S_\epsilon$ [equation (37)] is contracting.

The left-hand side of equation (45) quantifies the amount of dissipation that is "seen" at each node of the network, which vanishes for lossless couplings. The right-hand side accounts both for the strength of the coupling between each pair of oscillators, through the weights, and for the connectedness of the graph, $\lambda_2$ being the algebraic connectivity[67]. Under our assumptions [$G_u$ is connected, couplings are strictly increasing on $R(\boldsymbol{\gamma})$], the right-hand side of equation (45) is necessarily positive. For couplings with sufficiently low dissipation, equation (45) is then satisfied, which, combined with Theorem 4, yields the following corollary.

**Corollary 6.** If equation (45) is satisfied, then there is at most a unique synchronous state of equation (2) in each winding cell. The number of synchronous states is then bounded from above by the number of winding cells.

Theorem 5 and Corollary 6 give a rigorous, even though conservative, sufficient condition for at most uniqueness of synchronous states in each winding cell. However, computing the eigenvalues of the odd weighted Laplacian can be time-consuming. We, therefore, propose some lower bounds on $\lambda_2(L_f)$ in the Supplementary Information (Prop. S2) that are state-independent and may ease the verification of equation (45). The bounds are adapted from standard results of algebraic graph theory.

## Data availability
The data used in this article have been deposited on the `DFNSolver`[41] repository: https://doi.org/10.5281/zenodo.5899408.

## Code availability
The codes created for this article have been deposited on the `DFNSolver`[41] repository: https://doi.org/10.5281/zenodo.5899408.

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

## Acknowledgements
R.D. was supported by the Swiss National Science Foundation, under grant number P400P2_194359. This work was partly supported by AFOSR grant FA9550-22-1-0059.

## Author contributions
All three authors designed the research; R.D. and S.J. derived the mathematical results; R.D. did the simulations; R.D. and F.B. analyzed the simulations and wrote the paper.

## Competing interests
The authors declare no competing interests.
