## [Peer Review File · Nature Communications]

Title: Multistability and Anomalies in Oscillator Models of Lossy Power GridsREVIEWER COMMENTS

Reviewer #1 (Remarks to the Author):

In this manuscript the authors show a correspondence between stable synchronous states in dissipatively coupled oscillators, and the winding partition of their state space.

In this framework they found that increased dissipation can lead to more robust and more stable systems. Moreover, for a small amount of dissipation, the authors show that, exactly as in lossless networks, there is at most one solution of the dissipative flow problem in each winding cell.

All findings are illustrated with the Kuramoto-Sakaguchi model, which represents a natural extension of the (lossless) Kuramoto model and at the same time it is a first order version of the lossy swing equations.

The presented results are a straight generalization of Refs. 20, 42 to dissipative systems.

Moreover not negligible results have been already presented in a manuscript not cited by the authors:

Philipp C. Böttcher, Dirk Witthaut and Leonardo Rydin Gorjão "Dynamic stability of electric power grids: Tracking the interplay of the network structure, transmission losses and voltage dynamics." arXiv preprint arXiv:1908.10083 (2019).

In Böttcher et al. analytical results and effective stability criteria are presented, focusing on the interplay of network structures and the local dynamics of synchronous machines. In particular the results are based on an extensive linear stability analysis of the third-order model for synchronous machines, comprising the classical power-swing equations and the voltage dynamics, thus using a more general and complex model with respect to the Kuramoto-Sakaguchi.

Therefore I believe that in the submitted manuscript are shown only minimal incremental advancements with respect to previous publications that do not justify a publication in Nature Communications

Reviewer #2 (Remarks to the Author):

I think this is a solid paper --- the theoretical component contains significant novelty, and the problem is well-motivated by specific applications. After having addressed the clarifying comments that I give below, I would be happy to see this paper published.

The theorems 2,3 are pretty standard in the Kuramoto world, so no surprises here, but it is quite useful to have the result in this broader context.

I'm not sure that I agree with the claim that is made in the paragraph that goes from p. 11 to p. 12, that λ_2 is positive. Or, at least, I'm not saying it's wrong but I don't think the present argument establishes this. Theorem 4 is a nice result and generalizes to networks in a powerful way.

One big issue I have, or might have, is really more of a philosophical one. The authors state three paradoxes in the Results section (and one of these even make the abstract). I'm not 100% sure that these are paradoxes, because the heuristic arguments that give their negation are more linguistic than mathematical. For example, if the authors had chosen to use "asymmetric" instead of "dissipative" here, then these paradoxes wouldn't exist. In any case, I think paradox is too strong --- perhaps "surprises" or something.

Also, the authors use the term dissipative in the introduction several times before ever saying what it means. It might be good to do this earlier. Perhaps even better: in addition to (1) being given as a prototypical example of a lossless model, why not give a concrete prototypical example of a dissipative network at this point in the paper, to give the reader a better idea of the type of model being explored here?

The figures in the paper are good and illustrative. I especially like Figure 3 as a visualization of the example.

Reviewer #3 (Remarks to the Author):

The article contains new results and it is well written.

Two items could be improved. There is a difference between the abstract where an algorithm is announced, which can calculate all equilibria. In contrast to this, there is a theorem for cyclic graphs and a theorem for acyclic graphs. With the latter the equilibrium can only be found in a cell. A better explanation how to find all equilibria should be included or if this is not possible in an easy way, then the statement in the abstract should be weakened.

A comment about the conservatism or sharpness of Formula (40) is desirably.

Manuscript NCOMMS-22-11433: Point-to-point answer

June 21, 2022

R. Delabays, S. Jafarpour, and F. Bullo

Reviewer #1:

In this manuscript the authors show a correspondence between stable synchronous states in dissipatively coupled oscillators, and the winding partition of their state space. In this framework they found that increased dissipation can lead to more robust and more stable systems. Moreover, for a small amount of dissipation, the authors show that, exactly as in lossless networks, there is at most one solution of the dissipative flow problem in each winding cell.

All findings are illustrated with the Kuramoto-Sakaguchi model, which represents a natural extension of the (lossless) Kuramoto model and at the same time it is a first order version of the lossy swing equations.

Thank you for this summary assessment, which we share. Your words accurately cover the content of our work.

The presented results are a straight generalization of Refs. 20, 42 to dissipative systems.

Indeed, our work builds up on previous results and approaches. We agree that, at a superficial level, the results presented may look as simple extensions of previous work. However, it is precisely one of the message of our manuscript to emphasize that upon careful inspection, this generalization is much less intuitive than it looks and then a rigorous treatment requires significantly more contribution than just applying the same tools all over again. First of all, as mentioned in our manuscript, a whole part of the previous analysis of diffusive oscillators networks does not apply because even with a small amount of dissipation, the system is not a gradient flow anymore. Furthermore, we illustrate the fact that the generalization to dissipative systems is not as straightforward as one can think at multiple points in the manuscript, e.g.,

- in the vectorial formulation of the ODEs, which we have not seen anywhere else in the literature;
- in the intrinsic asymmetry of the system, rendering any analysis much more intricate than in the lossless case (loss of symmetry in the linearization, loss of flow conservation, constant interplay between the undirected nature of the graph and the directed nature of the interactions). Whereas the lossless case was naturally expressed in a symmetric framework, introducing losses requires to work in a directed framework. It is one of the contributions of our work to show that analyzing lossy systems as a perturbation of lossless systems is both intricate to justify (mathematically speaking) but rigorously accurate;
- through the “anomalies” (former “paradoxes”), showing that intuition leads to spurious conclusions;

- in Figure 4 and the related discussion, we lay the grounds for a generalization of our analysis to more elaborate systems that lossy coupled oscillators, i.e., considering variable amplitudes for the oscillators.

Moreover not negligible results have been already presented in a manuscript not cited by the authors:

Philipp C. Böttcher, Dirk Witthaut and Leonardo Rydin Gorjão "Dynamic stability of electric power grids: Tracking the interplay of the network structure, transmission losses and voltage dynamics." arXiv preprint arXiv:1908.10083 (2019).

In Böttcher et al. analytical results and effective stability criteria are presented, focusing on the interplay of network structures and the local dynamics of synchronous machines. In particular the results are based on an extensive linear stability analysis of the third-order model for synchronous machines, comprising the classical power-swing equations and the voltage dynamics, thus using a more general and complex model with respect to the Kuramoto-Sakaguchi.

Indeed, there are some important connections between the paper by Böttcher et al. and ours. The fixed points/synchronous states that we are investigating are fundamentally the same. We acknowledge this in the new version of the manuscript.

However, Böttcher et al.'s paper deals with a rather different question. The paper provides a novel, elegant, and mathematically rigorous framework for studying the stability of lossy coupled oscillator with voltage amplitude dynamics. The authors leverage an argument of perturbation theory, showing that, a given fixed point is linearly stable in the lossy system if (i) it is linearly stable in the lossless systems and (ii) losses are small enough. Namely, they perform a linear stability analysis of fixed points of the generalized swing equations. This is an important contribution, but the analysis only covers local stability properties. Indeed, the authors do not say anything about existence, uniqueness, domain of attraction of the fixed points, or about the global phase portrait of such systems.

The aim of our work is to provide a global phase portrait of the system. In our work, we identify whole subsets of the state space where there is provably at most one stable equilibrium/synchronous state. Even though our analysis is not formally "global" because it does not cover the whole state space, it nevertheless goes far beyond a local stability analysis. We furthermore emphasize some key ingredients leading to multistability in dissipative networks of coupled oscillators and power grids.

Therefore I believe that in the submitted manuscript are shown only minimal incremental advancements with respect to previous publications that do not justify a publication in Nature Communications

While we respectfully disagree with the assessment, we would like to thank the Reviewer for their comments that have helped us sharpen our message and improve the manuscript. We hope that our answers and the new version of the manuscript will convince them that our manuscript is worth being published in Nature Communications.

Reviewer #2:

I think this is a solid paper --- the theoretical component contains significant novelty, and the problem is well-motivated by specific applications. After having addressed the clarifying comments that I give below, I would be happy to see this paper published.

The theorems 2,3 are pretty standard in the Kuramoto world, so no surprises here, but it is quite useful to have the result in this broader context.

We would like to thank the Reviewer for their comments and are glad that they judge our paper solid and novel. We try to address their comments in the revised version.

I'm not sure that I agree with the claim that is made in the paragraph that goes from p. 11 to p. 12, that λ_2 is positive. Or, at least, I'm not saying it's wrong but I don't think the present argument establishes this.

We thank the Reviewer for their questions that made us realize that we were maybe a bit too concise there, we clarified the sentence in the main text. The fact that the algebraic connectivity is positive is a standard results from Fiedler (1973). It follows from the the following facts:

- L_o is symmetric;
- L_o is diagonally dominant;
- L_o has positive diagonal, because we restrict oscillators' angles to the appropriate phase-cohesive set;
- Gershgorin's Disks Theorem.

Theorem 4 is a nice result and generalizes to networks in a powerful way.

One big issue I have, or might have, is really more of a philosophical one. The authors state three paradoxes in the Results section (and one of these even make the abstract). I'm not 100% sure that these are paradoxes, because the heuristic arguments that give their negation are more linguistic than mathematical. For example, if the authors had chosen to use "asymmetric" instead of "dissipative" here, then these paradoxes wouldn't exist. In any case, I think paradox is too strong --- perhaps "surprises" or something.

We thank the Reviewer for this observation. We may have been overtaken by our enthusiasm when we decided to denote these phenomena as “paradoxes”. We did not want to use the word “surprises” mainly for a matter of style, as it sounded a bit awkward in some occurrences. The revised manuscript adopts the word “anomalies” which we find appropriate. We hope the Reviewer will agree with us on this choice. We are ready to discuss such terminology further if useful.

To elaborate on the fact that referring to “asymmetric” rather than “dissipative” coupling would make the “paradoxes/anomalies” disappear, we think that this is only partially true. Indeed, referring to the coupling as being “asymmetric” does not directly suggest that “asymmetry” jeopardize synchrony. But it does not suggest that it promotes synchrony neither. However, interpreting “asymmetry” in the coupling as a “dissipation” (which is quite natural as we discuss in the manuscript), clearly gives some

intuition against synchrony. To the best of our knowledge, there is no other interpretation that would suggest that “coupling asymmetry” promotes synchrony. Therefore, we think that the intuition given by the “dissipative” interpretation is reasonable and useful.

Also, the authors use the term dissipative in the introduction several times before ever saying what it means. It might be good to do this earlier. Perhaps even better: in addition to (1) being given as a prototypical example of a lossless model, why not give a concrete prototypical example of a dissipative network at this point in the paper, to give the reader a better idea of the type of model being explored here?

Excellent observation and very appropriate suggestion! Even though we kept equation (1) as an illustration of the lossless systems, we added an illustration of lossy systems a few paragraphs thereafter. We also adapted the notation to make the introduction of dissipation as clear as possible.

The figures in the paper are good and illustrative. I especially like Figure 3 as a visualization of the example.

We are particularly happy that the Reviewer appreciates our figures, we put a lot of effort into making them clear and useful.

Reviewer #3:

The article contains new results and it is well written.

We thank the Reviewer for their nice comment.

Two items could be improved. There is a difference between the abstract where an algorithm is announced, which can calculate all equilibria. In contrast to this, there is a theorem for cyclic graphs and a theorem for acyclic graphs. With the latter the equilibrium can only be found in a cell. A better Explanation how to find all equilibria should be included or if this is not possible in an easy way, then the statement in the abstract should be weakened.

Indeed, the algorithm we mention is not explicitly given in the manuscript, but provided in a repository accompanying the manuscript (Ref. 41). We clarified this point in the abstract.

A comment about the conservatism or sharpness of Formula (40) is desirably.

We extended the related paragraph in the Discussion section.

REVIEWERS' COMMENTS

Reviewer #1 (Remarks to the Author):

The authors have answered to my questions in a satisfactory way